

# Comprehensive evaluation of high rocky slope safety through an integrated analytic hierarchy process and extension matter model

H.Z. Su[1,2], M. Yang[2,3], Z.P. Wen[4]

[1]State Key Laboratory of Hydrology-Water Resources and Hydraulic Engineering, Hohai University, Nanjing 210098, China
5  [2]College of Water Conservancy and Hydropower Engineering, Hohai University, Nanjing 210098, China
[3]National Engineering Research Center of Water Resources Efficient Utilization and Engineering Safety, Nanjing 210098, China
[4]Dept. of Computer Engineering, Nanjing Institute of Technology, Nanjing 211167, China

*Correspondence to*: H.Z. Su (su_huaizhi@hhu.edu.cn)

10  **Abstract.** High rocky slope is an open complex giant system with contradiction among different influencing factors and coexistence of qualitative and quantitative information. This study presents a comprehensive intelligent evaluation method of high rocky slope safety by an integrated analytic hierarchy process, extension matter element model and entropy-weight to assess the safety behavior of the high rocky slope. The proposed intelligent evaluation integrates subjective judgments derived from the analytic hierarchy process with the extension matter model and entropy-weight into a multiple indexes 15  dynamic safety evaluation approach. A combining subjective and objective comprehensive evaluation process, a more objective manner through avoiding subjective effects on the weights and a qualitative safety assessment and quantitative safety amount are presented in the proposed method. The detailed computational procedures were also provided to illustrate the integration process of the above methods. Safety analysis of one high rocky slope is conducted to illustrate that this approach can adequately handle the inherent imprecision and contradiction of the human decision-making process and 20  provide the flexibility and robustness needed for the decision maker to better monitor the safety behavior of high rocky slope. This study was the first application of proposed integrated evaluation method to safety assessment of high rocky slope, which also indicated that it can also be applied to other similar problems.

## 1 Introduction

Nowadays, human beings are facing many serious environmental and natural geological disasters, accompanying the 25  massive construction of important projects (Su et al., 2013a). More and more high and steep rocky slopes are faced and threaten people's life and property and the safety of the whole projects. Those slopes safety behavior is needed to be urgently analyzed. From the engineering point of view, a high rocky slope is a complex open system and the factors affecting its stability are various and capricious which have strong chaotic characteristic. The complexity and uncertainty of those factors leads to the contradiction among different factors and the coexistence of qualitative and quantitative information, which may



seriously affect the security assessment of slopes. Therefore, it is very difficult and important to analyze the safety assessment of high rocky slopes.

From the viewpoint of expert systems, the final evaluations for the safety of one high rocky slope vary from person to person according to their different professional backgrounds, viewpoints, conditions of understanding the project, etc. Accordingly, the problem of safety evaluation of high rocky slope includes multiple criteria, both qualitative and quantitative features, and the problem is considered formulating as a multiple criteria decision making problem. The paper is to express the artificial reasoning in mathematical forms to mimic the human behavior, which belongs to expert systems. Analytic hierarchy process - Matter element analysis - Entropy weight model presents a fundamental approach to mathematically express human beings' inference and interpretation. The prevailing side of the proposed method in solving the multiple criteria decision making problem is its capability of more objective and intelligent assessment in mathematical form combining detected influencing factors without the need of human involvement and interference.

Many researches mainly focus on one specific aspect of slopes' safety and lack the identification of the whole security of high rocky slopes (Malkawi et al., 2000; PrSilva et al., 2008). Although there are many evaluation methods, the research and applications are still few and superficial. Traditional methods always tend to cause contradictions among affecting factors and the coincidence of qualitative and quantitative information, resulting in the uncertainty and incompletement.

In recent years, we have seen the development of the finite element analysis, the fuzzy mathematic method and the artificial neural network method. They all conduct the assessment through combining qualitative and quantitative information and the factors concerned are basically changeless (Su et al., 2013b). For the rocky slope safety evaluation, some advanced methods, such as, geographic information system and ground-based radar have been attempted for slope safety evaluation and warning (Dai and Lee, 2002; Bozzano et al., 2010; Casagli et al., 2010). However, existing interpretation methods typically applied one single index (e.g. surface deformation or rainfall) as a predictor and hence revealed only one aspect of slope performance (Ermini et al., 2005). The paper develops a method covering a multiple monitoring indexes and influencing parameters.

For the current one year researches from 2014 to 2015, the analysis method mainly focuses on the Bayesian network, Finite element analysis, Genetic Algorithm, Artificial neural network, Digital elevation model, Statistics method, Evolutionary approach, etc.(Abdalla et al., 2015; Jamsawang et al., 2015; Fuchs et al., 2014; Bahsan et al., 2014; Peng et al., 2014; Garg and Tai, 2014; Ji et al., 2015). The disadvantages of present rocky slope safety evaluation method are about one or certain factor, local safety analysis, single qualitative evaluation, single quantitative calculation, changeable results, more subjective analysis and so on. A holistic assessment of the slope safety state may not be achieved. Besides, a large portion of monitoring data is often not utilized in these methods.

The geological conditions of high rocky slopes are changeable and the outer environment also changes time after time. The evaluation methods and theories still have large limitations and defects which need to be further developed (Pantelidis, 2011). Slope safety evaluation using multiple sources of monitoring information may be more reasonable, and is the topic of the present research. For present studies, the whole comprehensive evaluation analysis combing subjective and objective



factors and presenting qualitative evaluation conclusion and quantitative evaluation amount is desired. Based on above reviews, an integrated approach is proposed avoiding above disadvantages and making up present research missing of high rocky slope safety evaluation.

The objective of this study is to propose a methodology for identifying and delineating high rocky slope safety using
integrated Analytic hierarchy process (AHP) - Matter element analysis (MEA) - Entropy weight method (EWM). Firstly, the previously related researches are reviewed briefly. Secondly, the multiple criteria decision making (MCDM) process of high rocky slope safety evaluation is computerized and a MCDM evaluation system of three layers-four criteria-fifteen sub-criteria is built by the AHP. Thirdly, MEA is used to determine the factor importance sequence to avoid subjective judgment as much as possible and sutra field, controlled field and the matrices of matter-element is set up from each layer and the
importance ranking of each factor is finally determined based on the single index correlation, comprehensive correlation and grade variable eigenvalue. The initial weight follows with the multiple comparison and judgment principle and matrix. A second correction with the EWM is conducted to reduce the decision maker's subjectivity on the largest level to improve the scientific nature and accuracy of the evaluation indexes. Lastly, an example of one slope is presented to illustrate the proposed methodology.

## 15 2. Literature Review

### 2.1. Multiple criteria decision making (MCDM)

The problems of MCDM are common occurrences in all kinds of areas while MCDM refers to making decisions in the presence of multiple, usually conflicting, criteria (Hwang and Yoon, 1981). Since the problem of safety evaluation of high rocky slope includes multiple criteria, both qualitative and quantitative features, the problem is considered formulating as a
20 MCDM problem. The subjective information and objective information coexist in MCDM. Thus, the subjective and objective approach is promoted to solve MCDM problems. For example, Kersuliene et al (2010) proposed the new step-wise weight assessment ratio analysis method to allow a more thoughtful application of existing MCDM analytic methodologies. Ginevicius (2011) adopted a new method, factor relationship for the MCDM problems. To solve the inherent uncertainty and imprecision of the MCDM, a fuzzy assessment approach was introduced, combing the concept of entropy and interval
normalization procedure in a fuzzy AHP (Ozkir and Demirel, 2012). Zolfani et al (2013) combined the stepwise weight assessment ratio analysis and weighted aggregated sum product assessment methods to solve the MCDM problem. A technical analysis results in the theoretical foundations of different MCDM methods and advantages and weaknesses among them were full shown by Larichev and Olson (2001), Belton and Stewart (2002), Hashemkhani (2013) and Figueira et al. (2005).

In addition, a fundamental problem of MCDM is to derive weight for a set of activities according to importance. Thus, scaling numbers from 1 to 9 were introduced to the weights of the elements in each level of the hierarchy with respect to an element of the next higher level in MCDM (Saaty, 1977). Weights based on a subjective approach reflect subjective

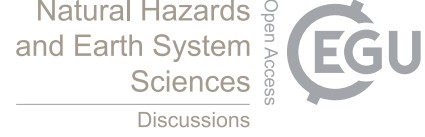



judgements from one person and objective weights obtained by mathematical methods are based on the analysis of the initial data. Both of them are not perfect and an integrated approach could be most appropriate for determining the weights of the attributes. Therefore, a integrated approach was introduced to determine attribute weights based on subjective approach, weighted least square method and objective approach, complicated multiple attribute decision making (Ma et al., 1999). As

the quantitative criteria were precisely defined, the developed quantitative multiple criteria decision making method can be successfully applied (Ustinovichius et al., 2007). In terms of qualitative and quantitative criteria, a decision-making technique was applied including Technique for Order Preference by Similarity to Ideal Solution, Simple Additive Weighting, and mixed methods as well as AHP and Entropy methods for defining importance of weights of the attributes (Karami and Johansson, 2014).

## 2.2. Analytic Hierarchy Process (AHP)

According to advantages of applied techniques, it is expected that AHP method can provide closer results to real decisions made (Saaty, 2003) and Saaty's AHP is a popular MCDM technique and widely used. Thus, AHP method is firstly chosen as the basic analysis approch in this paper. AHP was proposed by Thomas L. Saaty (Saaty, 1972; Shannon, 1948; Saaty, 1986; Saaty, 1988; Saaty, 1994; Vaidya and Kumar, 2006) and reflected the natural behavior of human thinking. Its major

innovation is the introduction of pair-wise comparisons. The pair-wise comparison technique represents a theoretically founded approach to compute weights representing the relative importance of criteria. Therefore, AHP has been considered as a very useful tool for dealing with complex MCDM problems by the international scientific community.

In the civil engineering, especially for the situations with incomplete and limited data, vague environments and great uncertainty inherent, AHP was developed to combine a risk assessment model (Abdul-Rahman et al., 2013). Li et al. (2013)

analyzed the inconsistent comparison matrix in AHP and proposed an improved AHP to improve comparison matrix consistency by using a sorting and ranking methodology. To manage crisis on the reconstruction of the damaged areas, the AHP method for weighing the important criteria and a novel complex proportional assessment of alternatives with gray relations for evaluating the alternatives, were successfully applied (Bitarafan et al., 2012). Li and Zou (2011) proposed a fuzzy AHP method to deal with the public private partnerships in infrastructure projects. Eshtehardian et al. (2013) applied

the analytical network process and AHP methods to select appropriate supplier in the construction and civil engineering companies and achieved a satisfied result. A MCDM approach was presented based on AHP for pedestrian zone selection under lack of quantitative data (Sayyadi and Awasthi, 2013). However, the models considered present some difficulties for application and are far from being perfect requiring further analysis (Triantaphyllou, 2000).

## 2.3. Matter Element Analysis (MEA)

Based on previous reviews and summaries, the AHP is proved to be a well-known decision support tool used for complex MCDM problems by providing a multi-level hierarchical structure. However, it is also indicated that using a single



mathematical method is difficult to finish the final decision making process for some drawbacks of the AHP. Thus, this paper adopts an integrated method to evaluate the high rocky slope safety.

During the AHP process, the index importance rule is influenced seriously by the artificial disturbance and it is difficult to establish a reasonable and precise index importance sequence. The incompatible problem is very evident. To solve above problems, Matter Element Analysis (MEA) method based on extension theory is introduced and primarily used to study the problem of incompatibility (Cai 1983).

Matter element analysis is a new theory to find out the regular patterns of incompatible problems. It can better reflect the change law of the objective things and help do qualitative analysis and quantitative calculation. Extenics theory differs from the fuzziness concept described in the fuzzy mathematic and the certain concept from the classical mathematic. It concentrates on the changeability of the research objects and further describes the pros and cons from the qualitative level to the quantitative level. It can be used for solving multiple parameters evaluation problem by formalizing the problem and establishing the corresponding matter element.

While solving the MCDM problem, three factors of MEA, namely events, features, values, are used to describe and represent the objects to be assessed and it is called matter element. It can better reflect the change law of the objective things and help do qualitative analysis and quantitative calculation which is very suitable for the decision maker to give a more precise safety assessment. Based on the matter element method, Cheng (2001) gave a practical study case of blasting classification of rock and Tang et al. (2009) evaluated soil nutrient of the ecological fragile region. Also, there are other detailed applications by matter element method, such as, weather forecasting and evaluating the financial security (Feng and Hong, 2014; Li et al., 2014). To expand the application of MEA, Wang and Lu (2015) proposed a predicting rockburst intensity model based on the fuzzy matter element theory.

This paper is based on the matter element concept fusing the evaluation objects, indexes and index values and the extension nature of the matter element is used to assess the whole safety of the high rocky slope.

## 2.4. Entropy weight method (EWM)

However, the safety evaluation weight of the slope founded on the AHP-extension matter model might be subjective, which might affect the objective analysis about its safety. In addition, for the integrated AHP and MEA, a minor factor can have a great influence on ranking the alternatives. Hence, entropy approach is introduced for determining the objective weight (Ustinovichius, 2001). AHP depends on experts to different extents and contains strong subjectivity. Slight weights differences can dramatically change the order of the alternative preference. Consequently, the appropriate weight for each criterion in MCDM is critical to achieve a precise qualitative evaluation and quantitative safety amount. An objective weight correction method is an important guarantee.

As we all know, Entropy is the one of the most important concepts in social science, physics, and information theory. The entropy weight method (EWM) is used to determine the weights of influence indexes as a better way to avoid subjective influence. EWM can be quantified and simulated to represent the objective information contained in a MCDM system and





the subjective information possessed by a decision maker. In the literature of information theory, the concept of Entropy, was firstly introduced by Shannon (1948). Shannon's Entropy served as a useful method for evaluating data structures and patterns. Then, EWM was introduced in determination of criteria weights for MCDM problems (Shemshadi et al, 2011; Ye, 2010; Kildiene et al, 2011). Ge et al (2013) established a fuzzy optimization model based on EWM to select typical flood
hydrograph in the design flood and this model was relatively better to design flood computation compared with the traditional typical flood hydrograph. European country management capabilities within the construction sector in the time of crisis were investigated applying a proposed method determined via entropy method (Susinskas et al., 2011). Assessment and selection of appropriate solutions for occupational safety was classified as MCDM problem and EWM was used to determine relative significances of evaluation criteria (Dejus and Antucheviciene, 2013).

**2.5. Integrated method**

A single method effect is weak for the solution of MCDM problem and they may be dangerously inaccurate for complex decision problems. An integrated approach brings a comprehensive and objective result based on previous reviews. Thus, the integrated approach is an effective evaluation method. AHP and EWM are often used to implement the safety evaluation of hydraulic or civil engineering (Kok et al. 2009; Rahman et al., 2013). Tavana and Hatami-Marbini (2011) developed a
MCDM framework based on AHP and EWM for the human spaceflight mission planning. For the selection of the best material for the tool holder used in hard milling, Caliskan et al. (2013) applied the AHP and EWM to confirm criteria weighting so that a new decision model was developed. For the pursuing the goal of selecting the best transportation investment project portfolio, a fuzzy assessment approach was presented by utilizing the concept of entropy and fuzzy AHP (Ozkir and Demirel, 2012).
This paper adopts an effective integrated method to assess the safety of high rocky slope, the AHP-MEA-EWM approach, for above reviews. It sets up the comprehensive safety evaluation model of multi-layer and multi-objective for high rocky slopes from the aspects of geological condition, engineering condition, external environment and internal and external monitoring behavior. AHP is firstly used to construct a multiple layers and multiple indexes MCDM framework. Then, the importance ranking of all influential factors of high rocky slopes is determined based on the single index correlative degree,
comprehensive correlative degree and grade variable eigenvalue of the matter element matrix. The initial weight is obtained based on above analyses and a second correction with the entropy weight method is made. Finally, the quantitative safety amount and qualitative safety level are determined by overlapping application of integrated methods and recursive calculation from the criteria level to the goal level.

**3. Multi-level and multi-index evaluation system of high rocky slope safety**

Single index of a complex high rocky slope is not enough for evaluating its safety. And, the stability of the high rocky slope is obviously influenced by its inner characteristics and outer environment. Therefore, it is necessary to set up a multi-level





and multi-index evaluation structure. Each level plays a dominant role with its adjacent subsequent level covering multiple indices and each forms a layer-by-layer dominant relationship from top to bottom. Following above method, the hierarchy of an open-cut construction of a high rocky slope has been established as shown in Fig. 1. The highest level (Level 1) of the hierarchy represents the overall goal evaluating the high rocky slope safety. The second level (Level 2) is of all concerns in the high rocky slope safety, consisting of geological conditions, engineering conditions, outer environment and inner and outside monitoring. The general criteria constitute the lowest level (Level 3).

According to the specification and national standard of the water conservancy and hydropower slope engineering about the rocky slope safety classification and learning from the researches about the ranking standard from the correlated scholars, the differences of different areas and different projects are comprehensively considered (Wu et al. 2009; Zhao et al. 2011). Combined with numbers of engineering cases, the safety grade of high rocky slope can be divided into 5 levels shown as Table 1, namely Ⅰ means very stable, Ⅱ means stable, Ⅲ means basically stable, Ⅳ means unstable and Ⅴ means extremely unstable.

In order to eliminate the shortcomings of incommensurability brought by the dimension and dimensional unit for evaluation index values under different factors, the dimensionless method is used to process the evaluation indexes on different grade-levels. For the index values with the larger the better tendency, hence the range formula applied here is

$$c'_{ij} = \left( c_{ij} - c_{i\,\max} \right) \Big/ \left( c_{i\,\max} - c_{i\,\min} \right) \tag{1}$$

For the smaller the better index value under different levels, the range formula is

$$c'_{ij} = \left( c_{i\,\max} - c_{ij} \right) \Big/ \left( c_{i\,\max} - c_{i\,\min} \right) \tag{2}$$

where $c_{i\max}$ and $c_{i\min}$ are the maximum and minimum indicators at different levels of one indicator. $c_{ij}$ and $c'_{ij}$ are the initial index value and the final result calculated by self-developed algorithm of range method under one indicator level. The dimensionless result is shown in Table 1.

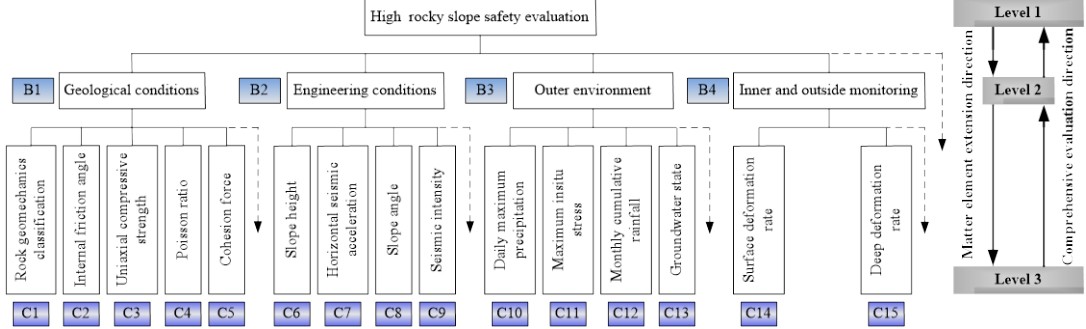

**Figure 1.** Safety evaluation hierarchy of high rocky slope.





**Table1.** Value rank of evaluation index under different grades.

| Index | Normal index value [Dimensionless index value] | | | | |
|---|---|---|---|---|---|
| | I | II | III | IV | V |
| Rock geomechanics classification | 80~100 | 60~80 | 40~60 | 20~40 | 0~20 |
| | [0.80~1.00] | [0.60~0.80] | [0.40~0.60] | [0.20~0.40] | [0.00~0.20] |
| Internal friction angle (°) | 60~90 | 50~60 | 39~50 | 27~39 | 0~27 |
| | [0.67~1.00] | [0.56~0.67] | [0.43~0.56] | [0.30~0.43] | [0.00~0.30] |
| Uniaxial compressive strength (MPa) | 150~200 | 125~150 | 90~125 | 40~90 | 10~40 |
| | [0.74~1.00] | [0.61~0.74] | [0.42~0.61] | [0.16~0.42] | [0.00~0.16] |
| Poisson ratio | 0~0.20 | 0.20~0.25 | 0.25~0.30 | 0.30~0.35 | 0.35~0.50 |
| | [0.60~1.00] | [0.50~0.60] | [0.40~0.50] | [0.30~0.40] | [0.00~0.30] |
| Cohesion force (MPa) | 0.22~0.32 | 0.12~0.22 | 0.08~0.12 | 0.05~0.08 | 0.00~0.05 |
| | [0.69~1.00] | [0.38~0.69] | [0.25~0.38] | [0.16~0.25] | [0.00~0.16] |
| Slope height (m) | 0~75 | 75~175 | 175~300 | 300~500 | 500~1000 |
| | [0.93~1.00] | [0.83~0.93] | [0.70~0.83] | [0.50~0.70] | [0.00~0.50] |
| Horizontal seismic acceleration (g) | 0.00~0.05 | 0.05~0.10 | 0.10~0.15 | 0.15~0.20 | 0.20~0.40 |
| | [0.88~1.00] | [0.75~0.88] | [0.63~0.75] | [0.50~0.63] | [0.00~0.50] |
| Slope angle (°) | 0~10 | 10~20 | 20~30 | 30~40 | 40~90 |
| | [0.89~1.00] | [0.78~0.89] | [0.67~0.78] | [0.56~0.67] | [0.00~0.56] |
| Seismic intensity | 0~2 | 2~4 | 4~6 | 6~8 | 8~12 |
| | [0.83~1.00] | [0.67~0.83] | [0.50~0.67] | [0.33~0.50] | [0.00~0.33] |
| Daily maximum precipitation (mm) | 0~20 | 20~40 | 40~60 | 60~100 | 100~150 |
| | [0.87~1.00] | [0.73~0.87] | [0.60~0.73] | [0.33~0.60] | [0.00~0.33] |
| The maximum in-situ stress (MPa) | 0~2 | 2~8 | 8~14 | 14~20 | 20~25 |
| | [0.92~1.00] | [0.68~0.92] | [0.44~0.68] | [0.20~0.44] | [0.00~0.20] |
| Monthly cumulative rainfall (mm) | 0~50 | 50~100 | 100~150 | 150~250 | 250~300 |
| | [0.83~1.00] | [0.67~0.83] | [0.50~0.67] | [0.17~0.50] | [0.00~0.17] |
| Groundwater state (L•min$^{-1}$• (10m)$^{-1}$) | 0~25 | 25~50 | 50~100 | 100~125 | 125~150 |
| | [0.83~1.00] | [0.67~0.83] | [0.33~0.67] | [0.17~0.33] | [0.00~0.17] |
| Surface deformation rate (mm•d$^{-1}$) | 0~2 | 2~3 | 3~5 | 5~8 | 8~10 |
| | [0.80~1.00] | [0.70~0.80] | [0.50~0.70] | [0.20~0.50] | [0.00~0.20] |
| Deep deformation rate (mm•d$^{-1}$) | 0.00~0.20 | 0.20~0.30 | 0.30~0.50 | 0.50~1.00 | 1.00~2.00 |
| | [0.90~1.00] | [0.85~0.90] | [0.75~0.80] | [0.50~0.75] | [0.00~0.50] |

## 4. The proposed integrated method evaluating high rocky slope safety

### 4.1. Importance identification of evaluation indexes on high rocky slope

Comprehensive stability evaluation in terms of high rocky slope is based on the extensibility of matter-element for stability evaluation. Its core is to determine the extension domain and the matter-element transformation of factors is the implementation means in the extension domain. For extenics, the matter-element is the logic cell. The stability of high rocky slope is called event $N$. The feature of high rocky slope stability called $C$ including the geological conditions, the engineering conditions, the external environment and the internal and external monitoring behavior. Eigenvalue of each factor in the every level is called the magnitude $V$. The ordered triple $R=\{N,C,V\}$ is known as the basic element to be described, also



named as matter-element. Introduction of matter element provides a viable tool for solving formalization of stability comprehensive evaluation questions of high rocky slopes.

Extension evaluation will be established including every level basing on the matter-element extensibility. Sutra field, controlled field and the matrices of matter-element for appraising will be set up from each layer. The importance ranking of each factor is finally determined in the corresponding level basing on the single index correlation, the comprehensive correlation and the grade variable eigenvalue. By that, it can avoid subjective judgments as far as possible and provide an important basis for the improved AHP.

(1) Matter element

For the evaluation level matter-element, evaluation event $N_{0k}$ are composition elements of evaluation level. The indexes composing evaluation level are matter-element feature $C_i$ including $C_1$, $C_2$, …, $C_n$ and the index value is called matter-element feature of the magnitude $V_{0ik}=<a_{0ik}, b_{0ik}>$. $R_{0k}=(N_{0k}, C_i, V_{0ik})$ means evaluation level matter-element, its sutra field matrix is expressed as

$$R_{0k}=\left(N_{0k},C_i,V_{0ik}\right)=\begin{bmatrix} N_{0k} & C_1 & V_{01k} \\ & C_2 & V_{02k} \\ & C_3 & V_{03k} \\ & \vdots & \\ & C_n & V_{0nk} \end{bmatrix}=\begin{bmatrix} N_{0k} & C_1 & <a_{01k},b_{01k}> \\ & C_2 & <a_{02k},b_{02k}> \\ & C_3 & <a_{03k},b_{03k}> \\ & \vdots & \\ & C_n & <a_{0nk},b_{0nk}> \end{bmatrix} \tag{3}$$

Its controlled field can be expressed as

$$R_p=\left(P,C_i,V_{pi}\right)=\begin{bmatrix} P & C_1 & V_{p1} \\ & C_2 & V_{p2} \\ & C_2 & V_{p3} \\ & \vdots & \\ & C_n & V_{pn} \end{bmatrix}=\begin{bmatrix} P & C_1 & <a_{p1},b_{p1}> \\ & C_2 & <a_{p2},b_{p2}> \\ & C_2 & <a_{p3},b_{p3}> \\ & \vdots & \\ & C_n & <a_{pn},b_{pn}> \end{bmatrix} \tag{4}$$

where $R_p$ means controlled field matter-element; $P$ means the composition of evaluation level elements and $V_{pi}=<a_{pi}, b_{pi}>$ means the maximum magnitude range of the controlled field matter-element and $<a_{0ik}, b_{0ik}> \subset <a_{pi}, b_{pi}>$, $(i=1,2,…,n)$.

Furthermore the matrices of matter-element for appraising can be expressed as

$$R_t=\left(N_t,C_t,V_t\right)=\begin{bmatrix} N_m & C_1 & d_1 \\ & C_2 & d_2 \\ & C_2 & d_3 \\ & \vdots & \\ & C_n & d_n \end{bmatrix} \tag{5}$$



where $R_t$ means the matter-element of high rocky slope safety to be evaluated; $N_t$ means elements of evaluation level to be evaluated; $C_t$ means evaluating planning index of evaluation level elements; $V_t = d_i$ ($i=1,2\ldots n$) means the actual value of $C_m$.

(2) Correlation function determination

Based on the extenics theory, the calculation results of simple correlation function of evaluation indexes play an important role in determining the weight coefficient. The expression is

$$K_t(V_m) = \begin{cases} \dfrac{2(d_m - a_{0mt})}{b_{0mt} - a_{0mt}} & d_m < \dfrac{a_{0mt} + b_{0mt}}{2} \\[3mm] \dfrac{2(b_{0mt} - d_m)}{b_{0mt} - a_{0mt}} & d_m \geq \dfrac{a_{0mt} + b_{0mt}}{2} \end{cases} \tag{6}$$

The expression of elementary correlation function is

$$K_t(V_m) = \begin{cases} \dfrac{\rho(d_m, V_{0mt})}{\rho(d_m, V_{pm}) - \rho(d_m, V_{0mt})} & \rho(d_m, V_{pt}) - \rho(d_m, V_{0mt}) \neq 0 \\[3mm] -\rho(d_m, V_{0mt}) - 1 & \rho(d_m, V_{pt}) - \rho(d_m, V_{0mt}) = 0 \end{cases} \tag{7}$$

where

$$\rho(d_m, V_{0mt}) = \left| d_m - \frac{1}{2}(a_{0mt} + b_{0mt}) \right| - \frac{1}{2}(b_{0mt} - a_{0mt})$$

$$= \begin{cases} a_{0mt} - d_m, & d_m < \frac{1}{2}(a_{0mt} + b_{0mt}) \\[3mm] d_m - b_{0mt}, & d_m \geq \frac{1}{2}(a_{0mt} + b_{0mt}) \end{cases} \tag{8}$$

$$\rho(d_m, V_{pm}) = \left| d_m - \frac{1}{2}(a_{pm} + b_{pm}) \right| - \frac{1}{2}(b_{pm} - a_{pm})$$

$$= \begin{cases} a_{pm} - d_m, & d_m < \frac{1}{2}(a_{pm} + b_{pm}) \\[3mm] d_m - b_{pm}, & d_m \geq \frac{1}{2}(a_{pm} + b_{pm}) \end{cases} \tag{9}$$

Above correlation function formula, $K_t(V_m)$ expresses simple correlation function value for the $t$-th classification grade of $d_m$, which represents the $m$-th ($m=1,2,\ldots,n$) indicator in the evaluating planning matter element. $a_{0mt}$ and $b_{0mt}$ represent the minimum and maximum values of the index grades respectively. $\rho(d_m, V_{0mt})$ and $\rho(d_m, V_{pm})$ represent the interval distance between evaluating planning index value and $<a_{0ik}, b_{0ik}>$, $<a_{pi}, b_{pi}>$ respectively.

(3) Importance ranking of each factor in the same index level



The evaluation index maximum correlation function value $K_{m\max}$ and the associated stability grade $t_{mK\max}$ ($m=1,2,\ldots,n$) are calculated via simple or elementary dependent function. The matrix is composed as follows.

$$T = \left\{ t_{1K_{\max}}, t_{2K_{\max}}, \ldots, t_{nK_{\max}} \right\} \tag{10}$$

Then, the maximum grade of $t$-grade matrix can be determined as

$$T' = \left\{ \max(T) \right\} = \left\{ t_{1t}', t_{2K_{\max}}', \ldots, t_{xK_{\max}}' \right\} \tag{11}$$

where $x$ represents the $x$-th indicator in the maximum $t$ grade in index level, and the maximum value does not exceed $n$. $t_{xK\max}'$ represents the grade value of the $x$-th indicator in the maximum $t$ grade matrix.

In the maximum grade matrix $t$, the index $x$ corresponding to the maximum grade value is the most important factor in the index level followed by the index corresponding to the second maximum value. The rest can be deduced by analogy. In this way, the index sequence of importance at the maximum grade can be determined.

Then the second largest grade is considered, building the second largest grade matrix and determining the sequence importance of every factor based on the similar process above. At last, the importance sequence of different factors in every indicator layer can finally be determined.

## 4.2. Weight determination of evaluation indexes on high rocky slope

According to the scale of multiple comparison and judgment principle, A.L.Satty applied fuzzy mathematics theory to establish multiple comparison scale system (Table 2) (Xu, 2013; Chowdary et al., 2013). Considering the factor importance sequence identification in index level, comparative judgment matrix of adjacent upper level is finally established. The judgment matrix is solved by square root method and the consistency check is carried on. And then, the factor weight vector of index level is determined. Index weight is maximized objective correction based on comprehensive weight determining method of the combination of subjective and objective factors of multi-factor entropy-weight model. Ultimately the weighted value is determined.

(1) Construct judgment matrix

According to the understanding and preliminary analysis of high rocky slope, the elements involved in the slope evaluation system is arranged hierarchically by nature, namely the establishment of the overall goal level, criteria level and sub-criteria level.

Judgment matrix is the basic information of the AHP. It is also the basis to calculate the relative importance and level single sequencing. The method takes a factor of upper level as the criterion and builds multiple comparison judgment matrix based on the above conclusions about the factor importance sequence and the standard table of multiple comparison in Table 2. The result is listed in Table 3. The every factor of judging matrix satisfies the following relations.

$b_{ij}=1/b_{ji}$, $b_{jj}=1$ ($i, j=1, 2\ldots n$) \hfill (12)



**Table 2.** Standard table of multiple comparison.

| Scale value $a_{ij}$ | Implication |
| --- | --- |
| 1 | $B_i$ and $B_j$ are equally important |
| 3 | $B_i$ is slightly more important than $B_j$ |
| 5 | $B_i$ is obviously more important than $B_j$ |
| 7 | Compared with the $B_j$, $B_i$ is very important |
| 9 | Compared with the $B_j$, $B_i$ is extremely important |
| 1/3 | $B_j$ is slightly more important than $B_i$ |
| 1/5 | $B_j$ is obviously more important than $B_i$ |
| 1/7 | Compared with the $B_i$, $B_j$ is very important |
| 1/9 | Compared with the $B_i$, $B_j$ is extremely important |
| 2,4,6,8 | The importance of $B_i$ compared with $B_j$ is between the corresponding degree above |

**Table 3.** Judgment matrix of evaluation system on high slope safety.

| $A$ | $B_1$ | $B_2$ | ... | $B_j$ | ... | $B_n$ |
| --- | --- | --- | --- | --- | --- | --- |
| $B_1$ | 1 | $b_{12}$ | ... | $b_{1j}$ | ... | $b_{1n}$ |
| $B_2$ | $b_{21}$ | 1 | ... | $b_{2j}$ | ... | $b_{2n}$ |
| $\vdots$ | $\vdots$ | $\vdots$ | $\vdots$ | $\vdots$ | $\vdots$ | $\vdots$ |
| $B_i$ | $b_{i1}$ | $b_{i2}$ | ... | $b_{ij}$ | ... | $b_{in}$ |
| $\vdots$ | $\vdots$ | $\vdots$ | $\vdots$ | $\vdots$ | $\vdots$ | $\vdots$ |
| $B_n$ | $b_{n1}$ | $b_{n2}$ | ... | $b_{nj}$ | ... | 1 |

(2) Calculate the eigenvalue and eigenvectors

$MW=\lambda_{\max}W$, $W$ and $\lambda_{\max}$ represent the eigenvector and eigenvalue of the judgment matrix. The square root method is applied while it is generally approximated solution.

The product $E_i$ ($i=1,2…,n$) of every row element can be calculated by multiplying the elements in the judgment matrix $M$ by row.

$$E_i = \prod_{j=1}^{n} b_{ij} \tag{13}$$

Calculate the $n$-th root of $E_i$ of each row ($n$ is the order of matrix) as follows.

$$\overline{E}_i = \sqrt[n]{E_i} \tag{14}$$

Take $\overline{E}_i$ into regularization processing as follows.

$$E'_i = \overline{E}_i \bigg/ \sum_{k=1}^{n} \overline{E}_k \tag{15}$$

$E'_i$ is called the weight vector of the matrix.

Estimate the maximum characteristic value of judgment matrix as follows.

$$\lambda_{\max} = \sum_{k=1}^{n} \left[ \left(ME'\right)_k \big/ \left(nE'_k\right) \right] \tag{16}$$



The consistency check index *CI* is introduced to make the result better consistent with the actual situation. Consistency check formula is as follow.

$$CR=CI/RI \qquad (17)$$

where $CI=(\lambda_{max}-n)/(n-1)$. The mean random consistency index, *RI*, is different along with the change of the matrix order. The result is shown in Table 4. Judgment matrix is introduced. When $CR<0.1$, the consistency of judgment matrix is acceptable. Otherwise, it needs to adjust the judgment matrix until the consistency test meets the requirements. Especially, when $CR<0.01$, the consistency of the matrix is a satisfactory result. When $CR=0$, it is a complete consistency check.

(3) Determine and correct the weight vector

Through the above process, the judgment matrix meets the requirements of consistency check. The vector, *E´*, is the weight vector of judgment matrix constituted by influential factors of high slope. According to the nature of the entropy, actual information and subjective information of decision makers are quantified and synthesized to build up the multi-objective decision entropy weight model from which we can learn how decision makers influence the index weight. Then the weight indicator of AHP by entropy-weight method is corrected to make the index weight vector more reasonable and accurate. Factor importance ranking method built by entropy-weight method and matter-element extension theory reduce the subjective judgment as a maximization level, making the results more accurate and objective.

The following formula is adopted to determine the entropy value $e_i$ of the *i*-th indicator.

$$e_i = -\frac{1}{\ln(n)}\sum_{j=1}^{n}\left(\frac{c'_{ji}}{\sum_{j=1}^{n}c'_{ji}}\right)\ln\left(\frac{c'_{ji}}{\sum_{j=1}^{n}c'_{ji}}\right) \qquad (18)$$

where *n* is the slope judgment matrix order. $c'_{ij}$ is the slope impact factor index value which has experienced the range treatment.

The difference coefficient $g_i$ of the *i*-th indicator is calculated as follows.

$$g_i = (1-e_i)\bigg/\left(n-\sum_{i=1}^{n}e_i\right) \qquad (19)$$

where $0\leq g_i\leq 1$.

The information weight value $w_i$ is determined as follows.

$$w_i = g_i\bigg/\sum_{i=1}^{n}g_i \qquad (20)$$

The index weight $k_i$ of AHP is corrected by $w_i$, and eventually the synthesis weight $s_i$ is determined as follows.

$$s_i = w_i k_i\bigg/\sum_{i=1}^{n}w_i k_i \qquad (21)$$




**Table 4.** Different order value of RI.

| The order of judgment matrix | Consistency index $RI$ | The order of judgment matrix | Consistency index $RI$ |
|---|---|---|---|
| 1 | 0.00 | 8 | 1.41 |
| 2 | 0.00 | 9 | 1.45 |
| 3 | 0.58 | 10 | 1.49 |
| 4 | 0.90 | 11 | 1.52 |
| 5 | 1.12 | 12 | 1.54 |
| 6 | 1.24 | 13 | 1.56 |
| 7 | 1.32 | 14 | 1.58 |

## 4.3. High rocky slope safety state determination

With the help of single factor correlation function value $K_t(V_m)$ and the weight vector $s$, it can confirm the extension correlation $K_t(O)$ of the evaluation planning slope $O$ about the grade of $t$.

$$K_t(O) = \sum_{i=1}^{n} s_i K_t(V_i) \tag{22}$$

where $\sum_{i=1}^{n} s_i = 1$, $n$ is the matrix dimension.

With extension relativity, the safety state of the high rocky slope can eventually be obtained, namely

$$K_t(O) = \left\{ k_{11}(O), k_{22}(O), .\ k_{ti}(O).\ k_{ss}(O) \right\} \tag{23}$$

where $t$ is the standard grade; $i$ is the element numbering of the vector $K_t(O)$ at the grade $t$, and the elements maximum value in this vector are recorded as

$$k_{it0}(O) = \max \left( \left\{ k_{11}(O), k_{22}(O), .\ k_{ti}(O).\ k_{ss}(O) \right\} \right) \tag{24}$$

The stability grade of high slope is $t_0$. The calculation formula for eigenvalue of classification grade of the evaluation planning index can be expressed as follow.

$$\overline{k}_t(O) = \frac{\left[ k_t(O) - \min K_t(O) \right]}{\left[ \max(K_t(O)) - \min(K_t(O)) \right]} \tag{25}$$

where

$$t' = \sum_{t=1}^{s} t\overline{k}_t(O) \bigg/ \sum_{t=1}^{s} \overline{k}_t(O) \tag{26}$$

Fig. 2 shows the implementing process of high rocky slope safety evaluation with the proposed approach.



## 5. Case study

Integrating the aforesaid principles and methods, the calculation procedure for the evaluation of high rocky slope has been compiled in this part. This procedure is designed with the consideration of achieving the real-time dynamic evaluation and internal and external monitoring timeliness. The AHP-MEA-EWM is adopted during the calculation. One high rocky slope in China is used as a typical research project analyzed under the multi-factor evaluation system, as shown in Table 5.

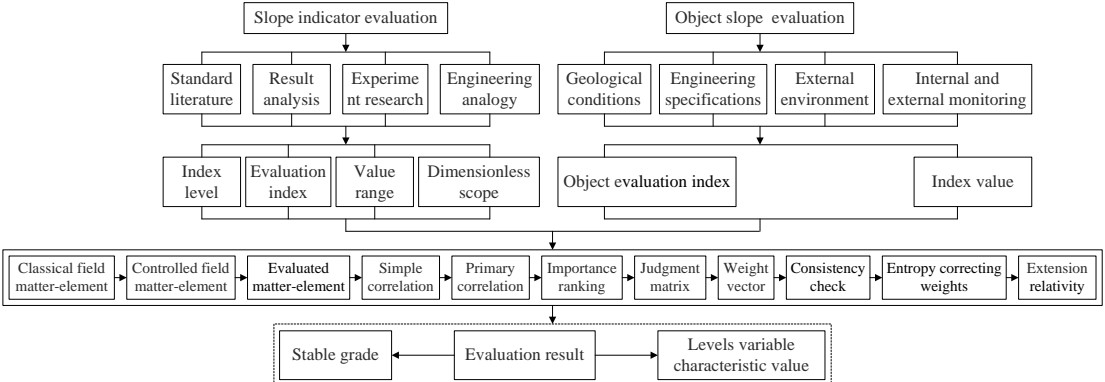

**Figure 2.** Evaluation flowchart of high rocky slope safety.

**Table 5.** Safety evaluation index value of high rocky slope.

| Level 2 | Level 3 | Values | Dimensionless values |
|---|---|---|---|
| Geological conditions B1 | Rock geomechanics classification(RMR) | 30.00 | 0.30 |
| | Internal friction angle (°) | 19.30 | 0.21 |
| | Uniaxial compressive strength (MPa) | 110.00 | 0.53 |
| | Poisson ratio | 0.35 | 0.30 |
| | Cohesive force (MPa) | 0.08 | 0.25 |
| Engineering conditions B2 | Slope height (m) | 750.00 | 0.25 |
| | Horizontal seismic acceleration (g) | 0.07 | 0.83 |
| | Slope angle (°) | 43.00 | 0.52 |
| | Seismic intensity | 5.00 | 0.58 |
| External environment B3 | Daily maximum precipitation (mm) | 54.60 | 0.64 |
| | Maximum in-situ stress (MPa) | 23.50 | 0.06 |
| | Monthly cumulative rainfall (mm) | 68.40 | 0.77 |
| | Groundwater state (L•min$^{-1}$•(10m)$^{-1}$) | 75.00 | 0.50 |
| Internal and external monitoring B4 | Surface deformation rate (mm•d$^{-1}$) | 9.90 | 0.010 |
| | Deep deformation rate (mm•d$^{-1}$) | 1.63 | 0.19 |



## 5.1. Influencing index importance analysis

On the basic ideas above, the matter-element extension matrix of the index level should be constructed firstly to determine the element importance sorting. Then simple correlation function values and elementary dependent function values can be are calculated through establishing its sutra field, controlled field and the matrices of matter-element for appraising. In order to identifying the index importance ranking, the improved AHP is used to construct the judgment matrix and the initial weight vector is made after a consistency inspection. At last, the EWM based on multi-objective decision is as a mean of secondary revision for initial weight vector to build the extension relativity. The final safety state is determined for the high rocky slope and the dimensionless parameters in this project are shown in Table 5.

This article takes the index evaluation system under Geological conditions (B1) criteria in Fig. 1 as an example to do concrete analysis. And other indexes can be analyzed by the similar analysis process. For the stability classification, sutra field ($R_{01} \sim R_{05}$), controlled field ($R_p$), and the evaluating planning matter element ($R_0$) are respectively shown as follows.

$$R_{01} = \begin{bmatrix} N_{01} & C_1 & <0.80,1.00> \\ & C_2 & <0.67,1.00> \\ & C_3 & <0.74,1.00> \\ & C_4 & <0.60,1.00> \\ & C_5 & <0.69,1.00> \end{bmatrix}, R_{02} = \begin{bmatrix} N_{02} & C_1 & <0.60,0.80> \\ & C_2 & <0.56,0.67> \\ & C_3 & <0.61,0.74> \\ & C_4 & <0.50,0.60> \\ & C_5 & <0.38,0.69> \end{bmatrix}, R_{03} = \begin{bmatrix} N_{03} & C_1 & <0.40,0.60> \\ & C_2 & <0.43,0.56> \\ & C_3 & <0.42,0.61> \\ & C_4 & <0.40,0.50> \\ & C_5 & <0.25,0.38> \end{bmatrix},$$

$$R_{04} = \begin{bmatrix} N_{04} & C_1 & <0.20,0.40> \\ & C_2 & <0.30,0.43> \\ & C_3 & <0.16,0.42> \\ & C_4 & <0.30,0.40> \\ & C_5 & <0.16,0.25> \end{bmatrix}, R_{05} = \begin{bmatrix} N_{05} & C_1 & <0.00,0.20> \\ & C_2 & <0.00,0.30> \\ & C_3 & <0.00,0.16> \\ & C_4 & <0.00,0.30> \\ & C_5 & <0.00,0.16> \end{bmatrix}, R_p = \begin{bmatrix} P & C_1 & <0.00,1.00> \\ & C_2 & <0.00,1.00> \\ & C_3 & <0.00,1.00> \\ & C_4 & <0.00,1.00> \\ & C_5 & <0.00,1.00> \end{bmatrix},$$

$$R_0 = \begin{bmatrix} N_0 & C_1 & 0.30 \\ & C_2 & 0.21 \\ & C_3 & 0.53 \\ & C_4 & 0.30 \\ & C_5 & 0.25 \end{bmatrix}$$

(27)

According to Eqs. (6) and (7), the simple correlation function and elementary correlation function values of $R_{01}$ are determined based on calculating correlation procedure developed in this paper. The calculation results are listed in Tables 6 and 7.

The result of importance ranking of different factors according to the above methods is $C_2$, $C_4$, $C_1$, $C_5$, $C_3$. Based on the same method, the results of importance order of other indexes in Level 3 are $C_8$, $C_6$, $C_9$, $C_7$; $C_{11}$, $C_{13}$, $C_{10}$, $C_{12}$; $C_{15}$, $C_{14}$. According to the importance order and Table 2, the judgment matrix of $B_1$-C is shown in Table 8.



$$R_{B1-C} = \begin{pmatrix} 1 & 1/3 & 3 & 1/2 & 2 \\ 3 & 1 & 5 & 2 & 4 \\ 1/3 & 1/5 & 1 & 1/4 & 1/2 \\ 2 & 1/2 & 4 & 1 & 3 \\ 1/2 & 1/4 & 2 & 1/3 & 1 \end{pmatrix} \tag{28}$$

Based on the procedure of calculating weights and eigenvalues developed in this paper, the obtained weight result is (0.16, 0.42, 0.06, 0.26, 0.10) with 5.07 as its eigenvalue. By consistency test ($CR$=0.015<0.1), the judgment matrix has a satisfactory consistency which shows that the judgment matrix and the weight are reliable. Based on the weight amendment

procedure, the result of corrected weight vector is (0.18, 0.55, 0.044, 0.14, 0.083). According to Eqs (22)-(26) and program developed in this paper, the extension relativity of each level is calculated (-0.63,-0.53,-0.35,-0.083, 0.33), and the eigenvalue of the classification grade is 4.25.

Similarly, it is the same way for other index layers to determine the related results including the weight vector, final weight result after amendment, eigenvalue, extension relativity and the eigenvalue of the classification grade, which are as

follows: B2-C with its weight vector result is (0.28, 0.10, 0.47, 0.16) and eigenvalue of 4.03 has a satisfactory consistency in consistency test ($CR$=0.0096<0.01). The weight after amendment is (0.27, 0.11, 0.44, 0.18) and the extension relativity in each level is (-0.48, -0.33, -0.27, -0.26, -0.30). The eigenvalue of the classification grade is 3.11. B3-C is with its weight vector results as (0.16, 0.48, 0.88, 0.27) and eigenvalue is 4.00 with a satisfactory consistency in consistency test ($CR$=0.0034<0.01) and the weight vector after amendment is (0.17, 0.47, 0.091, 0.27). The extension relativity in each level

is (-0.63,-0.50,-0.27,-0.46, -0.69), and the eigenvalue of the classification grade is 2.85. B4-C is with its weight vector result of (0.25, 0.75) and eigenvalue of 1.99 and matrix order of 2 under a satisfactory consistency. The extension relativity in each level is (-0.84, -0.83, -0.81, -0.70, -0.86), and the eigenvalue of the classification grade is 3.57.

**Table 6.** Index level results of correlation function under geological conditions criteria with grade I.

| Evaluated index | Classic field interval | Controlled field interval | Evaluated matter element | Elementary dependent function | Simple correlation function |
|---|---|---|---|---|---|
| $C_1$ | <0.80,1.00> | <0.00,1.00> | 0.30 | -0.63 | -5.00 |
| $C_2$ | <0.67,1.00> | <0.00,1.00> | 0.21 | -0.69 | -2.79 |
| $C_3$ | <0.74,1.00> | <0.00,1.00> | 0.53 | -0.31 | -1.62 |
| $C_4$ | <0.60,1.00> | <0.00,1.00> | 0.30 | -0.50 | -1.50 |
| $C_5$ | <0.69,1.00> | <0.00,1.00> | 0.25 | -0.64 | -2.84 |

**Table 7.** The results of elementary dependent function under every grade.

| Index | I | II | III | IV | V |
|---|---|---|---|---|---|
| $C_1$ | -0.63 | -0.50 | -0.25 | 0.50 | -0.25 |
| $C_2$ | -0.69 | -0.63 | -0.51 | -0.30 | 0.75 |
| $C_3$ | -0.31 | -0.15 | 0.21 | -0.19 | -0.44 |
| $C_4$ | -0.50 | -0.40 | -0.25 | 0.00 | 0.00 |
| $C_5$ | -0.64 | -0.34 | 0.00 | 0.00 | -0.26 |





**Table 8.** Standard table of multiple comparison of $B_1$-C.

| B | $C_1$ | $C_2$ | $C_3$ | $C_4$ | $C_5$ |
|---|---|---|---|---|---|
| $C_1$ | 1 | 1/3 | 3 | 1/2 | 2 |
| $C_2$ | 3 | 1 | 5 | 2 | 4 |
| $C_3$ | 1/3 | 1/5 | 1 | 1/4 | 1/2 |
| $C_4$ | 2 | 1/2 | 4 | 1 | 3 |
| $C_5$ | 1/2 | 1/4 | 2 | 1/3 | 1 |

**Table 9.** Evaluation results of criteria level (Level 2).

| Index grade | Single index relativity | | | | Comprehensive correlation |
|---|---|---|---|---|---|
| | Geological conditions | Engineering conditions | External environment | Internal and external monitoring | |
| 1 | -0.81 | -0.53 | -0.46 | -0.64 | -0.56 |
| 2 | -0.75 | -0.37 | -0.28 | -0.52 | -0.41 |
| 3 | -0.63 | -0.06 | 0.07 | -0.29 | -0.13 |
| 4 | -0.25 | 0.06 | -0.07 | 0.43 | 0.068 |
| 5 | 0.50 | -0.32 | -0.35 | -0.23 | -0.22 |

## 5.2. Final safety evaluation

The analysis program of evaluation system is built on the theory of the integrated AHP-MEA-EWM. For the evaluation system of the overall goal level (Level 1), the analysis steps and ideas are consistent with criteria level (Level 2) and the classification eigenvalue of criteria level (Level 2) constructed above is (0, 1), (1, 2), (2, 3), (3, 4), (4, 5). Based on the calculated idea of this article, the result of single-index relation degree, integrated incidence degree and grade variable eigenvalue for safety evaluation of high rocky slope are shown in Table 9.

Based on above methods, the importance order is B3, B4, B2, B1. The initial value of the weight is (0.10, 0.16, 0.47, 0.28) and the value of the weight after amendment is (0.11, 0.18, 0.44, 0.27) and classification eigenvalue is 3.75. The safety state of grade slope is Ⅳ, which is fully consistent with the present situation that the high rocky slope under an unstable state with a larger and more unstable deformation. It is worth noting the result of multivariate analysis gives an alert of its precarious state. Based on the results of its internal and external monitoring, this safety evaluation system has provided an important theoretical basis for making rational decisions using a procedural algorithm to assess the security status of the slope timely and dynamically.

## 6. Conclusions

An integrated Analytic Hierarchy Process-Matter Element Analysis-Entropy Weight method for solving multiple criteria decision making problem has been proposed and applied to comprehensive safety assessment of high rocky slope. The proposed method integrates analytic hierarchy process method, matter element theory, and entropy weight method within a safety assessment framework. The specific multivariate computational procedures were provided to illustrate the integration process of the above methods. Its evaluation process involves factor input, self-evaluation, dynamic assessment and real-time



grade standards output.etc. The comprehensive assessment results demonstrate that, the level eigenvalue of one high rocky slope is 3.75, and its security status is Ⅳ fully consistent with the actual situation. Decision makers can conduct flexible and variable response programs for the high rocky slope. This study is the first application of the proposed method to safety assessment of high rocky slope.

Compared with the traditional method of solving MCDM problem, the proposed method not only can assess multi-criteria decision problems in a more objective manner through avoiding subjective effects on the weights, but also can simultaneously product the qualitative evaluation conclusion and quantitative evaluation amount. The effects of subjective errors on safety assessment of high rocky slope could be avoided at a large extent. The intelligent preliminary evaluation analysis system provides an important theoretical support for timely making scientific judgments based on the feedback of

safety state of high rocky slope. Thus, decision makers can make a more objective and flexible evaluation for the high rocky slope. Meanwhile, finding the "best" method solving the MCDM problem is an elusive goal that may never be reached. Limitations also exist in this research. For example, this method could not solve the safety state for fuzzy and uncertain factors. The imprecision of the human decision-making process might exist and all the analyses should be conducted before the evaluation of qualitative factors can be consistent as this method is still based on the AHP theory.

The proposed MCDM approach integrates subjective judgments derived from the AHP with MEA and EWM into an intelligent, preferable and subjective and objective and multiple criteria approach. The structured approach presented in this study has some obvious attractive features: (1) Comprehensive and flexible: The approach combines a comprehensive method covering the AHP, MEA and EWM technique concerning multiple factors, multiple criteria and multiple layers. The number of the all kinds of factors is not limited and this method is flexible for extension; (2) Structured and Analytical: The

miscellaneous factors are stratified into a hierarchy to simplify information input and provide a clear framework for the decision maker for so complex problem. The proposed method helps decision maker decompose a complex problem into manageable steps; (3) Qualitative and quantitative: The comprehensive approach presents a qualitative evaluation analysis and a quantitative evaluation amount, which can be listed for the decision maker to finish a more precise decision; (4) Subjective and objective: The subjective evaluation of a finite number of decision alternatives is allowed as the generic

nature of the proposed approach of this paper allow on a finite number influencing factors. The objective influencing factors importance of multiple layers can be obtained by MEA and an objective weight can be achieved by integrating the AHP and EWM; (5) Computational and intelligent: The proposed method is a mathematical and computational model which can be widely applicable to MCDM problem.

For future research, it can be conducted in the following directions: (1) the proposed method could be future developed

based on fuzzy theory for imprecise, ambiguous or unknown data; (2) a more objective evaluation guideline could be introduced to amend the subjective judgment of the AHP; (3) the proposed method could be applied to other MCDM problems, especially for the safety evaluation for other fields. (4) The matter element theory could be more widely applied in solving MCDM problem by integrating other methods.



## Acknowledgments

This research has been partially supported by National Natural Science Foundation of China (SN: 51579083, 41323001, 51139001, 51479054), Jiangsu Natural Science Foundation (SN: BK2012036), the Doctoral Program of Higher Education of China (SN: 20130094110010), Open Foundation of State Key Laboratory of Hydrology-Water Resources and Hydraulic Engineering (SN: 20145027612), the Fundamental Research Funds for the Central Universities (SN: 2015B25414), a Project Funded by the Priority Academic Program Development of Jiangsu Higher Education Institutions (SN: 3014-SYS1401).

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
