# Peer review of "Comprehensive evaluation of high rocky slope safety through an integrated analytic hierarchy process and extension matter model"

_Natural Hazards and Earth System Sciences, 2015_

## Referee Comment (RC1) · Anonymous Referee #1 · 3 Mar 2016

The paper proposes a approach implementing the safety evaluation for high rocky slope. Many factors influencing rocky slope safety are considered by building a multi-level and multi-index evaluation system. Some methods are integrated to fulfill the comprehensive evaluation of high rocky slope safety. The topic is overall within the major scopes of NHESS and may be of some interests to its general readers and, in particular, those specialized in civil engineering. The manuscript can be considered for publication with minor revisions.. 1.To make the original contribution be clear, it might be more suitable that the title of the manuscript is changed into "An approach using multi-factor combination to evaluate high rocky slope safety". 2. In Figure 1, the meaning of unique dotted arrow should be explained. 3. In Section "Case study", more

details might be added to better embody the present safety state of the high rocky slope. 4. The authors should check the final manuscript and avoid any grammatical error or syntax error.

---

## Referee Comment (RC2) · Anonymous Referee #2 · 3 Mar 2016

This manuscript studies some key problems and presents a case analysis of slope safety evaluation by some mathematical methods combined. The manuscript proposes an approach of comprehensive evaluation for slope safety. The proposed method consists of a systematic integration of several techniques, such as analytic hierarchy process, extension matter element and entropy. The proposed method has a high potential for other similar applications. Overall, I recommends the publication of this paper, after the suggested improvements properly addressed. <1>P7: In Equations (1) and (2), it will be more clear for the readers that one index is illustrated for 'the index values with the larger the better tendency' and 'the smaller the better index value under different levels', for instance, 'the index values with the larger the better tendency, such

as. . .' and 'the smaller the better index value under different levels, for example. . .'. <2>P14: Check Equations(23)∼(24). <3>P18: In Section '5.2 Final safety evaluation', 'The safety state of grade slope is âĚč, which is fully consistent with the present situation that the high rocky slope...'. However, the present situation of the high rocky slope is vague, which is advised to be supplied. <4>The paper described 'An integrated Analytic Hierarchy Process-Matter Element Analysis-Entropy Weight method for solving multiple criteria decision making problem has been proposed. . .', while the title only covered analytic hierarchy process and extension matter model.
* * *

---

## Referee Comment (RC3) · Anonymous Referee #3 · 8 Mar 2016

This paper focuses on the evaluation problem of slope safety. Some mathematical methods, namely Analytic Hierarchy Process, Matter Element Analysis, Information Entropy, are combined to build the evaluation index system, determine the index weight and establish the evaluation model. An actual engineering is appraised by the proposed method. The topic is overall within the major scopes of Natural Hazards and Earth System Sciences (NHESS). The paper presents an interesting approach. It implements the comprehensive analysis for the definite factors and the indefinite factors on slope safety. The proposed framework for analysis and evaluation of slope safety is practical. It is recommended that the authors consider the following points for clarification and completeness before their paper could be considered for publication: (1) The

**NHESSD**

title is ambiguous. (2) English needs to be checked in preparing the final manuscript. (3) P15: In Section '5. Case study', it is suggested to give more details on the current situation of analyzed slope.

---

## Referee Comment (RC4) · Anonymous Referee #1 · 21 Mar 2016

The paper was revised according to the comments. Also questions were answered. The revised manuscript can be considered for acceptance.

---

## Referee Comment (RC5) · Anonymous Referee #2 · 21 Mar 2016

The paper is upgraded and can be acceptable.

---

## Author Comment (AC1) · 21 Mar 2016

The paper proposes a approach implementing the safety evaluation for high rocky slope. Many factors influencing rocky slope safety are considered by building a multilevel and multi-index evaluation system. Some methods are integrated to fulfill the comprehensive evaluation of high rocky slope safety. The topic is overall within the major scopes of NHESS and may be of some interests to its general readers and, in particular, those specialized in civil engineering. The manuscript can be considered for publication with minor revisions. Response: Thank you very much for your evaluation and approval for our manuscript. We would like to express our great appreciation to you. We have tried our best to improve the manuscript.

[Figure]

1. To make the original contribution be clear, it might be more suitable that the title of the manuscript is changed into "An approach using multi-factor combination to evaluate high rocky slope safety". Response: According to your suggestion, the title of the manuscript has been changed into "An approach using multi-factor combination to evaluate high rocky slope safety".

2. In Figure 1, the meaning of unique dotted arrow should be explained. Response: Thank you very much for pointing it out. We have carefully explained the unique dotted arrow in Figure 1. There are lots of factors related to the rocky slope safety. Some of these factors are still unknown and a few factors are neglected for their ignorable influences to the rocky slope safety. Therefore, the dotted arrows are added to denote the unknown and ignorable factors for a more complete rocky slope safety evaluation process in Figure 1.

3. In Section "Case study", more details might be added to better embody the present safety state of the high rocky slope. Response: We are so sorry that we didn't express clearly in the original manuscript. According to your suggestion, more details have been added to better embody the present safety state of the high rocky slope in the Section "Case study" in Revised manuscript. The height of the unique high rocky slope is more than 700m and the wide is 50m~290m for an area of 115, 000m2. Larger deformations of the surface and inner rocks are caused for not good geological conditions under complex internal and external environments, such as rainfall, groundwater, and so on. The larger cumulative deformation velocity could reach to 3.5mm/d. At present, the maximum cumulative displacement amount is about 1500mm for one year. Therefore, the present rocky slope is unstable and its safety needs to be analyzed urgently by integrating multiple methods.

4. The authors should check the final manuscript and avoid any grammatical error or syntax error. Response: The authors have checked the final manuscript carefully for many times for voiding any grammatical error or syntax error. In addition, we have asked several colleagues who are skilled authors of English language papers to check

the paper.

Please also note the supplement to this comment:
http://www.nat-hazards-earth-syst-sci-discuss.net/nhess-2015-336/nhess-2015-336-AC1-supplement.pdf

**Supplement:**

**An approach using multi-factor combination to evaluate high rocky slope safety**

[revised manuscript text omitted]

---

## Author Comment (AC2) · 21 Mar 2016

This manuscript studies some key problems and presents a case analysis of slope safety evaluation by some mathematical methods combined. The manuscript proposes an approach of comprehensive evaluation for slope safety. The proposed method consists of a systematic integration of several techniques, such as analytic hierarchy process, extension matter element and entropy. The proposed method has a high potential for other similar applications. Overall, I recommends the publication of this paper, after the suggested improvements properly addressed. Response: Thank you very much for the interests and time given to review this paper. We really appreciate your evaluation and approval for our manuscript. Below you will find our point-by-point responses to

the comments.

<1>P7: In Equations (1) and (2), it will be more clear for the readers that one index is illustrated for 'the index values with the larger the better tendency' and 'the smaller the better index value under different levels', for instance, 'the index values with the larger the better tendency, such as: : :' and 'the smaller the better index value under different levels, for example: : :'. Response: Thank you very much for pointing it out. We have revised these expressions to be clearer for the readers. The revised sentences are listed as follows. 'the index values with the larger the better tendency, such as cohesion force, ' and 'the smaller the better index value under different levels, for example, slope height, '

<2>P14: Check Equations(23)∼(24). Response: Thank you very much for your suggestion. We have checked the Equations (23) - (24) in Microsoft Word and PDF. There is no problem in Microsoft Word. There might be some mistakes during conversion from Microsoft Word to PDF. We have produced a new PDF version which is no problem for the Equations (23) - (24).

<3>P18: In Section '5.2 Final safety evaluation', 'The safety state of grade slope is âĚč, which is fully consistent with the present situation that the high rocky slope...'. However, the present situation of the high rocky slope is vague, which is advised to be supplied. Response: Thank you very much for pointing it out and we agree with your precious comments. More details have been added to better embody the present situation of the high rocky slope to be clearer for the readers in Revised manuscript. The larger cumulative deformation velocity could reach to 3.5mm/d. At present, the maximum cumulative displacement amount is about 1500mm for one year. Therefore, the present rocky slope is unstable and its safety needs to be analyzed urgently by integrating multiple methods. Therefore, our conclusion is that the safety state calculated is fully consistent with the present situation that the high rocky slope. The whole process is clear, reasonable and complete.

<4>The paper described 'An integrated Analytic Hierarchy Process-Matter Element Analysis-Entropy Weight method for solving multiple criteria decision making problem has been proposed: : :', while the title only covered analytic hierarchy process and extension matter model. Response: Your suggestions are appreciative. The authors have revised the title and the new one is more generalized, which is 'An approach using multi-factor combination to evaluate high rocky slope safety'.

Please also note the supplement to this comment:
http://www.nat-hazards-earth-syst-sci-discuss.net/nhess-2015-336/nhess-2015-336-AC2-supplement.pdf

---

## Author Comment (AC3) · 21 Mar 2016

This paper focuses on the evaluation problem of slope safety. Some mathematical methods, namely Analytic Hierarchy Process, Matter Element Analysis, Information Entropy, are combined to build the evaluation index system, determine the index weight and establish the evaluation model. An actual engineering is appraised by the proposed method. The topic is overall within the major scopes of Natural Hazards and Earth System Sciences (NHESS). The paper presents an interesting approach. It implements the comprehensive analysis for the definite factors and the indefinite factors on slope safety. The proposed framework for analysis and evaluation of slope safety is practical. It is recommended that the authors consider the following points for clari-

fication and completeness before their paper could be considered for publication: Response: Thank you very much for your evaluation and approval for our manuscript. We would like to express our great appreciation to you. We have tried our best to improve the manuscript.

(1) The title is ambiguous. Response: We really appreciate your comment. We have revised the ambiguous title to be clear and accurate. The new title is 'An approach using multi-factor combination to evaluate high rocky slope safety'.

(2) English needs to be checked in preparing the final manuscript. Response: All the authors have checked the English carefully to void some errors. Meanwhile, we have asked several colleagues who are skilled authors of English language papers to check the paper.

(3) P15: In Section '5. Case study', it is suggested to give more details on the current situation of analyzed slope. Response: Your suggestions are appreciative. We have realized this problem as other reviewers also mentioned it. More details have been added to the current situation of analyzed slope in the manuscript. For instance, the maximum cumulative displacement amount is about 1500mm for one year. In the In Section5, Case study, we have given the detailed content for the current situation of analyzed slope. We hope that our modifications could be reasonable for your future review.

Please also note the supplement to this comment:
http://www.nat-hazards-earth-syst-sci-discuss.net/nhess-2015-336/nhess-2015-336-AC3-supplement.pdf
* * *
[Figure]

**Supplement:**

**An approach using multi-factor combination to evaluate high rocky slope safety**

[revised manuscript text omitted]

---

## Referee Comment (RC6) · Anonymous Referee #3 · 24 Mar 2016

I have carefully checked the satisfying revised manuscript and detailed responses for all the comments. The revised paper based on the comments is acceptable for the journal and it could be published for the current version
* * *